# Comparative application of testosterone undecanoate and/or testosterone propionate in induction of benign prostatic hyperplasia in Wistar rats

Jinho An[1,2], Hyunseok Kong[2,3]*

1 College of Pharmacy, Sahmyook University, Seoul, Korea, 2 College of PADAM Natural Material Research Institute, Sahmyook University, Seoul, Korea, 3 College of Animal Biotechnology & Resource, Sahmyook University, Seoul, Korea

* hskong0813@gmail.com

**Data Availability Statement:** All relevant data are within the paper and its Supporting Information files.

## Abstract

Testosterone undecanoate is a hormone agent with long-acting potential and is used for testosterone replacement therapy for hypogonadism. This study was designed to investigate application of testosterone undecanoate in maintaining high androgen levels for inducing benign prostatic hyperplasia more conveniently than that for testosterone propionate. We conducted two-part studies to determine the optimal dosage and dosing cycle for efficient and stable induction of benign prostatic hyperplasia using testosterone undecanoate. In the injection dosage substudy, single testosterone undecanoate dose (125, 250, 500, 750, or 1000 mg/kg body weight) was administered, and the optimal concentration was determined for 8weeks by measuring changes in testosterone, dihydrotestosterone, and 5-alpha reductase levels. And then, testosterone undecanoate was administered at the optimal dose at intervals of 1, 2, 3, or 4 weeks for 12weeks to induce benign prostatic hyperplasia. The injection dosage substudy showed dose-dependently higher and more stable levels of testosterone in groups administrated testosterone undecanoate than in groups administered testosterone propionate. In the injection cycle substudy, testosterone undecanoate-administered group stably maintained high levels of testosterone, dihydrotestosterone, and 5-alpha reductase compared with testosterone propionate-administered group for the same injection cycle; moreover, the prostate measurements, an important sign of benign prostatic hyperplasia, were significantly increased. Based on these two substudies, we determined the optimal conditions for inducing benign prostatic hyperplasia stably and more conveniently than that for testosterone propionate. This study suggests an extended application of testosterone undecanoate for inducing benign prostatic hyperplasia that can improve research reliability considering the half-life of testosterone as well as injection dosage and concentration.

**Funding:** This work was supported by the National Research Foundation of Korea (NRF) grant funded by the Korea government (MSIT) (No. 2019R1F1A1063286). There was no additional external funding received for this study. The funders had no role in study design, data collection and analysis, decision to publish, or preparation of the manuscript.

**Competing interests:** The authors have declared that no competing interests exist.

## Introduction

Testosterone is a steroid hormone derived from cholesterol, and it associated with various human body aspects such as sexual function, development, and systemic metabolism [1]. Testosterone preparations have been developed for use in clinical practice to treat hypogonadism and mimic physiological serum testosterone levels [2].

Since the half-life of testosterone for human application is only 10 min, an approach to develop testosterone formulations through esterification at the carbon 17 beta position is used (Fig 1) [3]. The first of these esters, testosterone propionate (T propionate), was marketed by Ciba and Schering in 1936, but had a short half-life of 1–2 days; however, the half-life of new testosterone formulations reported thereafter gradually increased.

Testosterone undecanoate (T undecanoate), mentioned as having favorable pharmacokinetics in the Development of methods of male contraception by world health organization (WHO), is a testosterone drug with the longest half-life, approved in more than 100 countries, including the approval by food and drug administration (FDA) in 2014; it is widely used in clinical trials for the treatment of gonad function [3–5]. Today exogenous testosterones were mainly used for testosterone replacement therapy, but the benign prostatic hyperplasia (BPH) and prostate cancer, the side effects of testosterone replacement therapy on prostate enlargement and related disorders, are commonly remained a concern for urologist [6]. Accordingly, we investigated the potential for extended application of T undecanoate, which is characterized by a long-acting effect in the body, beyond the treatment of hypogonadism.

We focused on BPH, one of the side effects of testosterone preparations. BPH is reported to increase by approximately 10–50% every 10 years in people above the age of 50; it is a common disease in middle-aged and elderly men that greatly affects their quality of life [7–10]. The pathophysiology of BPH reported various factors include genetic factors, androgens, oestrogens, insulin, growth factors, inflammation and stem cell, among these, androgens are regarded play a central role in the normal functional development of the prostate [11].

With increase in age, testosterone secretion decreases gradually, and testosterone is converted to dihydrotestosterone (DHT) in the presence of 5-alpha reductase; DHT acts as an endogenous ligand with high biological activity, being 2–3-fold and 15–30-fold more sensitive to androgen receptors than that of testosterone and adrenal androgens, respectively [12]. The production and accumulation of DHT in the prostate increases with age and cell proliferation that promotes the development of prostate tissue to induce BPH [13–15] and is mainly related to lower urinary tract symptoms such as frequent urination, dysuria, hematuria, urinary retention, and urinary tract infection [11,16,17].

In animal studies using rodents, exogenous testosterone was administered to castrated rats in which uncontrolled endogenous testosterone has been removed [18–27]. However, studies aimed to induce BPH with exogenous testosterone administration is challenging to maintain physiologically stable testosterone concentrations for a long period [28].

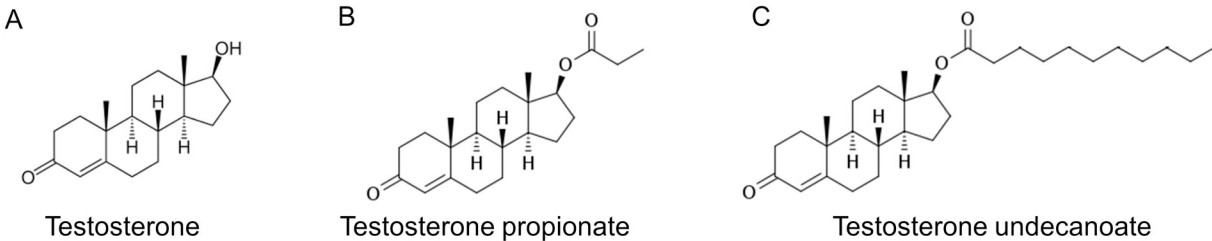

A Testosterone    B Testosterone propionate    C Testosterone undecanoate

**Fig 1.** Structure of testosterone (A), testosterone propionate (B), and testosterone undecanoate (C).

T propionate having the shortest half-life among testosterone preparations is still commonly used in most studies using rat for induction of BPH despite the considerable time and effort required to maintain high hormone levels in the body for a long period of time [18–27,29–31]. The dose of T propionate to induce BPH varied, and in most studies, daily testosterone treatment was employed to maintain high androgen levels in the body [18–27,29–31].

We hypothesized that application of T undecanoate, which has a long biological half-life, may contribute to a more convenient study of BPH via minimizing the number of injections compared to using T propionate that needs frequent injections for maintenance of stable and high androgen concentrations for long term.

## Materials and methods

### Materials

T undecanoate was purchased from Bayer Co., Ltd (Nebido®; Seoul, South Korea). T propionate was purchased from Avention Co., Ltd (Incheon, South Korea). Corn oil (Sigma-Aldrich, Korea) was used to dissolve the exogenous testosterone. Male Wistar Hannover rats, weighing 200–250 g, were purchased from Samtako Co., Ltd. (Osan, Republic of Korea) and were acclimatized for at least four weeks to a temperature of 20–24°C, relative humidity of 30–70%, and 12/12-h light/dark cycle. All rats were weighed once per week during the study. The animal studies were performed after receiving approval of the Institutional Animal Care and Use Committee (IACUC) of Sahmyook University (IACUC approval No. SYUIACUC2019-005, SYUIACUC2020-004).

### Hormone administration

At 12 weeks of age, following complete sexual and physical development, animals were randomly divided into different groups. Animals had ad libitum access to food and water, and administrated exogenous testosterone to induce BPH after castration or sham operation [24]. Hormonal injection was processed after operation of all animals immediately. Castration was conducted under anesthesia induced by 1:1 solution (0.1 ml/100g b.wt) of tiletamine/zolazepam (Zoletil 50; Virbac, France) and xylazine (Rompun; Bayer, Seoul, Republic of Korea), and the epididymal and testicular fatty tissues were removed before suturing [32,33]. Exogenous testosterone was administered subsequently. The study comprised two substudies: injection dosage substudy and injection cycle substudy. Both substudies included a normal control (NC) group that underwent sham operation, a castrated control (CC) group, and exogenous testosterone administration groups.

**The dosage substudy.** Rats in the administration group were castrated and administered a single subcutaneous injection of T undecanoate or T propionate at concentrations of 125, 250, 500, 750, or 1000 mg/kg body weight (b.wt) after their operation. The testosterone concentration was decided higher than 125 mg/kg b.wt, which is referred to previous study that report the higher level of serum testosterone compared to normal control after administration of T undecanoate [28]. Control groups were injected corn oil as a vehicle. Both control groups and 1000mg/kg b.wt group consisted of 5 rats, other group consisted of 6 rats at the beginning of the experiment, total 69 rats were used.

**The injection cycle substudy.** Rats in the administration group were castrated and injected T undecanoate or T propionate at a concentration of 125 mg/kg b.wt for 12 weeks, during which the administration groups received subcutaneous injections of exogenous testosterone at intervals of 1, 2, 3, or 4 weeks. Each group comprised four rats at the beginning of the experiment, total 40 rats were used.

## Blood sampling and hormonal analysis

Blood was collected from the jugular vein during the study period and from the abdominal vein at the end of the experiment. Serum was separated from the whole blood by centrifugation at 10,000 rpm at 4˚C for 5 min. Serum levels of testosterone, DHT, and 5-alpha reductase type 2 were determined using the corresponding rat ELISA kit (Abcam Cambridge, MA, USA; Cusabio, Barksdale, DE, USA; MyBioSource, San Diego, CA, USA) in accordance with the manufacturer's protocol. The detection range and sensitive of testosterone kit is 0.2 ng/ml to 16 ng/ml and 0.07 ng/ml; DHT kit is 10 pg/ml to 2000 pg/ml and 5pg/ml; 5-alpha reductase type 2 kit is 6 pg/ml to 1200 pg/ml and 3pg/ml.

## Prostate measurements

The prostate was weighed using an electronic scale (CUX220H; CAS Corporation, Seoul, Korea). The long (a) and short (b) dimensions of the prostate were measured using a digital caliper; the prostate volume (in cm3) was calculated using the following equation [33]:

$$\text{Prostate volume} = 1/2 \ (a \times b^2)$$

The prostate index was calculated from the body weight before sacrifice and the organ weight using the following formula [34]:

$$\text{Prostate index} = \text{organ wet weight}/\text{body weight} \times 100$$

## Histopathology

To analyze histopathological changes, the prostate tissue was fixed in 10% neutral formalin, infiltrated with paraffin, and embedded in paraffin blocks. The tissue was then cut into 4-μm-thick slices and stained with hematoxylin and eosin (H&E). Tissue specimens were examined using an optical microscope (Olympus, Tokyo, Japan) at 100× magnification to detect any changes in the thickness of the prostate epithelium [20,33].

## Statistical analysis

Data are presented as the mean ± standard deviation (SD). The significance of the mean differences among the NC, CC, and administration groups was determined using one-way analysis of variance (ANOVA), followed by Tukey-Kramer multiple comparison test in the injection dosage substudy and Dunnett's multiple comparison test in the injection cycle substudy. Three levels of significance were defined: $^*p < 0.05$, $^{**}p < 0.01$, and $^{***}p < 0.001$.

## Results

### Dosage substudy

**Changes in serum androgen levels.** To determine whether high androgen levels were maintained for inducing BPH, serum level changes of testosterone and DHT was measured for eight weeks after administration of different concentrations of T undecanoate and T propionate. Compared with the NC group, all T undecanoate-injected groups maintained significantly high level of testosterone for eight weeks, whereas all except 1000 mg/kg b.wt T propionate-injected groups maintained significantly high level of testosterone for 1–2 weeks only (Table 1). The level of DHT in 125 mg/kg b.wt and 250 mg/kg b.wt T undecanoate-injected groups were maintained significantly high for one week, and the level in the remaining T undecanoate-injected groups did not lower than that in the NC group for eight weeks. Moreover, 500–1000 mg/kg b.wt T undecanoate-injected groups maintained significantly high

level for eight weeks (Table 1). In contrast, 125–500 mg/kg b.wt, 750 mg/kg b.wt, and 1000 mg/kg b.wt T propionate-injected groups maintained significantly high level of DHT for only one, two, and three weeks, respectively (Table 1). Moreover, testosterone and DHT levels in T propionate-injected groups were decreased faster in a dose-dependent manner similar to the CC group than those in the T undecanoate-injected groups (Table 1).

**Changes in serum 5-alpha reductase.** The effect of exogenous testosterone on 5-alpha reductase, which converts testosterone to DHT, was evaluated by measuring its level at the end of the experiment. Significantly high level of 5-alpha-reductase was maintained in all T unde-canoate-injected groups, contrary to that in all T propionate-injected groups (Fig 2).

**Table 1. Changes in serum levels of testosterone and dihydrotestosterone in the dosage substudy.**

| Testosterone (ng/ml) | | 0week | 1week | 2week | 3week | 4week | 5week | 6week | 7week | 8week |
|---|---|---|---|---|---|---|---|---|---|---|
| NC | corn oil | 5.12±3.60 | 4.04±1.95 | 2.13±2.20 | 1.59±1.67 | 0.41±0.34 | 0.96±1.18 | 0.22±0.27 | 2.75±3.19 | 0.37±0.33 |
| | | | | | | | | | | |
| CC | corn oil | 2.12±2.08 | 0.02±0.02*** | 0.09±0.18 | 0.00±0.00 | 0.00±0.00 | 0.00±0.00 | 0.01±0.00 | 0.01±0.00 | 0.00±0.00 |
| T undecanoate | 125mg/kg | 6.01±5.00 | 17.15±4.53*** | 13.36±4.97** | 10.11±6.68 | 10.16±6.56* | 6.78±3.67* | 8.55±5.38** | 9.2±5.23* | 3.68±4.54** |
| | 250mg/kg | 5.99±5.40 | 13.76±6.51*** | 13.82±6.62** | 14.1±7.11* | 12.85±7.20** | 10.68±4.59*** | 10.55±5.36*** | 9.92±5.57* | 7.23±5.37*** |
| | 500mg/kg | 8.54±5.41 | 20.69±1.04*** | 19.54±1.50*** | 20.21±0.42*** | 19.72±0.83*** | 15.14±0.72*** | 14.18±1.48*** | 15.04±0.65*** | 13.26±0.8*** |
| | 750mg/kg | 9.30±6.43 | 20.49±1.32*** | 20.15±0.77*** | 20.49±1.07*** | 20.56±0.59*** | 15.32±0.50*** | 13.77±1.92*** | 15.21±0.43*** | 13.89±0.7*** |
| | 1000mg/kg | 9.64±5.25 | 20.34±0.76*** | 19.11±1.01*** | 17.98±3.29*** | 19.32±0.94*** | 14.46±1.06*** | 12.98±0.69*** | 15.12±0.47*** | 13.5±0.71*** |
| T propionate | 125mg/kg | 2.30±1.61 | 14.42±4.49*** | 5.95±7.07 | 2.21±3.89 | 0.26±0.63 | 0.13±0.30 | 0.84±1.45 | 0.01±0.01 | 0.00±0.00 |
| | 250mg/kg | 4.01±4.23 | 17.80±3.29*** | 1.57±2.46 | 0.29±0.67 | 0.00±0.00 | 0.00±0.00 | 0.01±0.00 | 0.00±0.00 | 0.00±0.00 |
| | 500mg/kg | 2.98±3.15 | 21.56±1.18*** | 13.85±8.31** | 11.77±8.47 | 5.65±6.38 | 2.68±3.49 | 1.59±2.64 | 0.06±0.14 | 0.01±0.01 |
| | 750mg/kg | 1.62±1.25 | 22.17±1.34*** | 18.59±4.38*** | 8.32±7.61 | 1.38±1.81 | 0.02±0.02 | 0.01±0.01 | 0.00±0.00 | 0.00±0.00 |
| | 1000mg/kg | 1.74±1.72 | 22.49±0.95*** | 21.43±0.88*** | 16.51±8.19** | 13.44±8.6*** | 8.22±5.20** | 9.08±6.73** | 5.06±6.26 | 5.11±5.72 |
| Dihydrotestosterone (ng/ml) | | | | | | | | | | |
| NC | corn oil | 1.52±0.3 | 1.29±0.22 | 1.33±0.23 | 1.18±0.23 | 0.98±0.15 | 0.48±0.12 | 0.38±0.10 | 0.51±0.22 | 0.4±0.12 |
| | | | | | | | | | | |
| CC | corn oil | 1.23±0.35 | 0.16±0.01*** | 0.49±0.30** | 0.28±0.04** | 0.39±0.12 | 0.11±0.03 | 0.06±0.03 | 0.04±0.02 | 0.07±0.02 |
| T undecanoate | 125mg/kg | 1.53±0.45 | 2.18±0.23*** | 1.83±0.15 | 1.74±0.53 | 1.39±0.65 | 0.65±0.38 | 0.79±0.37 | 0.79±0.39 | 0.57±0.38 |
| | 250mg/kg | 1.59±0.41 | 2.03±0.48*** | 1.83±0.31 | 1.93±0.29* | 1.65±0.36 | 1.01±0.21* | 0.89±0.23* | 0.85±0.26 | 0.78±0.31 |
| | 500mg/kg | 1.74±0.40 | 2.32±0.11*** | 2.10±0.04** | 2.13±0.03** | 2.04±0.03* | 1.16±0.02*** | 1.11±0.07*** | 1.14±0.03*** | 1.07±0.04*** |
| | 750mg/kg | 1.75±0.37 | 2.35±0.08*** | 2.14±0.03** | 2.14±0.02** | 2.05±0.08* | 1.14±0.05*** | 1.10±0.06*** | 1.14±0.04** | 1.11±0.01*** |
| | 1000mg/kg | 1.90±0.31 | 2.31±0.06*** | 2.10±0.02** | 2.13±0.02** | 2.05±0.02* | 1.05±0.12** | 1.07±0.12*** | 1.13±0.03** | 1.11±0.03*** |
| T propionate | 125mg/kg | 1.23±0.36 | 2.00±0.37*** | 1.27±0.66 | 0.96±0.63 | 0.54±0.40 | 0.17±0.18 | 0.24±0.26 | 0.11±0.06 | 0.02±0.01 |
| | 250mg/kg | 1.08±0.53 | 2.12±0.26*** | 1.08±0.43 | 0.75±0.42 | 0.17±0.07 | 0.13±0.07 | 0.08±0.03 | 0.06±0.03 | 0.03±0.01 |
| | 500mg/kg | 1.24±0.27 | 2.49±0.07*** | 1.86±0.34 | 1.77±0.40 | 1.04±0.86 | 0.41±0.42 | 0.30±0.31 | 0.12±0.16 | 0.06±0.06 |
| | 750mg/kg | 1.21±0.29 | 2.57±0.03*** | 2.07±0.12** | 1.60±0.35 | 0.93±0.53 | 0.23±0.13 | 0.07±0.04 | 0.06±0.02 | 0.05±0.02 |
| | 1000mg/kg | 1.24±0.23 | 2.57±0.05*** | 2.15±0.08** | 1.95±0.36* | 1.75±0.70 | 0.86±0.36 | 0.78±0.40 | 0.62±0.51 | 0.52±0.44 |

Changes in serum levels of testosterone and dihydrotestosterone (DHT), measured over a period of eight weeks after injecting testosterone undecanoate (T undecanoate) or testosterone propionate (T propionate). Treatment groups were injected T undecanoate or T propionate (125, 250, 500, 750, or 1000 mg/kg b.wt) after castration. Testosterone and DHT levels in serum were measured using commercial ELISA kit. Both control groups and 1000mg/kg b.wt group consisted of 5 rats, other group consisted of 6 rats at the beginning of the experiment. Values are expressed as the mean ± SD.

*p < 0.05

**p < 0.01

***p < 0.001 compared with normal control (NC). Statistical analysis was performed by ANOVA and Tukey-Kramer multiple comparison test.

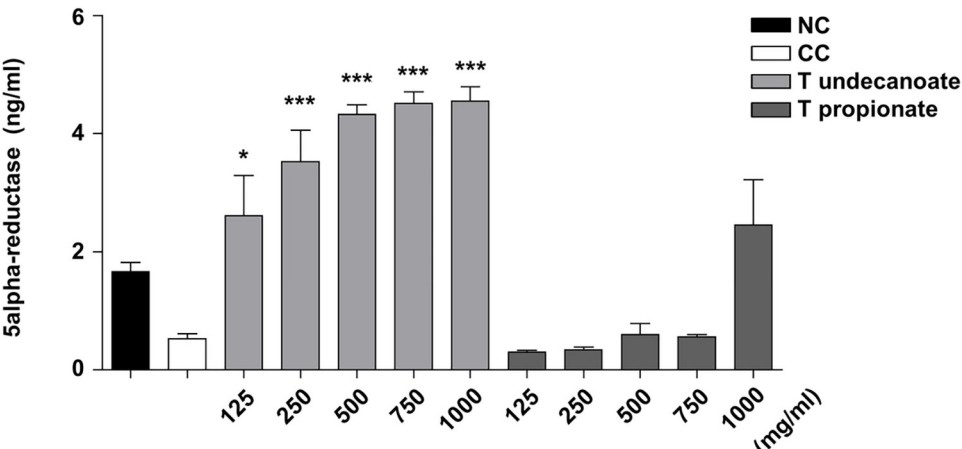

**Fig 2. The serum level of 5-alpha reductase.** The serum level of 5-alpha reductase after injecting testosterone undecanoate (T undecanoate) or testosterone propionate (T propionate). Treatment groups were injected T undecanoate or T propionate (125, 250, 500, 750, or 1000 mg/kg b.wt) after castration. Both control groups and 1000mg/kg b.wt group consisted of 5 rats, other group consisted of 6 rats at the beginning of the experiment. Values are expressed as the mean ± SD. *p < 0.05, **p < 0.01, ***p < 0.001 compared with normal control (NC). Statistical analysis was performed by ANOVA and Tukey-Kramer multiple comparison test.

**Changes in prostate measurements.** After evaluating the change in the serum levels of testosterone and DHT for eight weeks post administration of exogenous testosterone, prostate measurements were performed to determine its effect on prostate measurements following single administration after a long period. Compared with the NC group, prostate weight, prostate volume, and prostate index tend to increase in a dose-dependent manner in all T undecanoate-injected groups, with significant increase in prostate weight and prostate volume in 750 mg/kg b.wt and 1000 mg/kg b.wt T undecanoate-injected groups (Fig 3A–3C). In addition, compared with the NC group, prostate measurements in T undecanoate-injected groups did not decrease, whereas it did not increase and had low values in all T propionate-injected groups, except to 1000 mg/kg b.wt (Fig 3A–3C). We measured the organ weight to determine the degree of burden after administration for each concentration. The spleen weight in 1000 mg/kg b.wt T undecanoate-injected groups was significantly decreased, and the thymus weight decreased dose-dependently (S1 Table). The levels of AST, ALT and Creatinine, which indicate function of liver and kidney, were not shown remarkable change to determine dysfunction (S1 Table).

**Histomorphological changes in prostate tissue.** The main pathological feature of BPH is an increase in the epithelial thickness of the prostate tissue. Compared with the NC group, the epithelial thickness of the prostate tissue was significantly decreased in the CC group (Fig 3D and 3E). The epithelial thickness in 250–1000 mg/kg b.wt T undecanoate-injected groups was significantly increased in a dose-dependent manner, whereas it did not increase in all T propionate-injected groups (Fig 3D–3P).

## Injection cycle substudy

**Changes in serum androgen and 5-alpha reductase levels.** To confirm maintenance of high levels of androgen for inducing BPH, testosterone and DHT levels were measured. Compared with the NC group, testosterone level in the CC group was significant decreased, whereas T undecanoate injection maintained significantly high level of testosterone for 12 weeks at all time points (Table 2). In T propionate-injected groups, injection interval of one

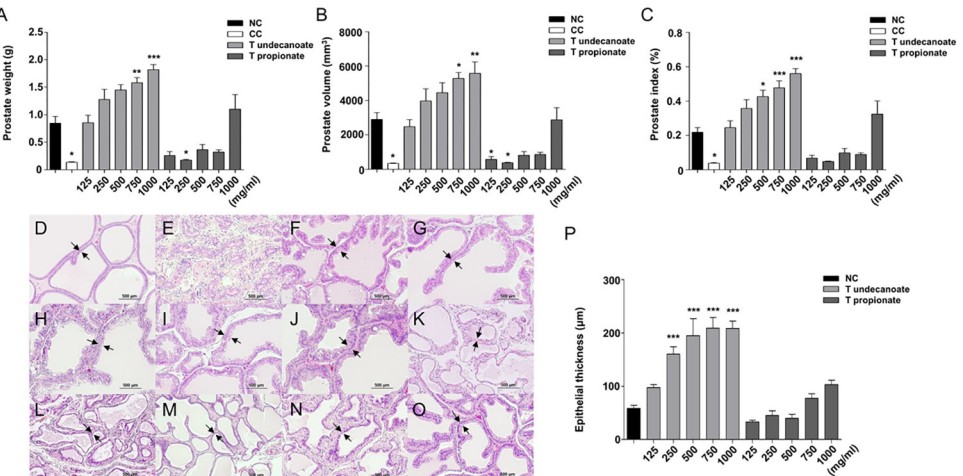

**Fig 3. Prostate measurements and histomorphological changes.** Prostate measurements and histomorphological changes in the prostate tissue after injecting testosterone undecanoate (T undecanoate) or testosterone propionate (T propionate). Treatment groups were injected T undecanoate or T propionate (125, 250, 500, 750, or 1000 mg/kg b.wt) after castration. Prostate weight in grams was measured (A). Prostate volume in cubic millimeters was measured using a caliper (B). Prostate index was calculated as (prostate weight/body weight) × 100 (C). Representative prostate tissue samples from each group were selected and stained with hematoxylin and eosin (H&E). Microscopic images of prostate samples are shown (100× magnification) from the following groups: Normal control (NC) (D); Castrated control (CC) (E); 125 mg/kg b.wt of T undecanoate (F); 250 mg/kg b.wt of T undecanoate (G); 500 mg/kg b.wt of T undecanoate (H); 750 mg/kg b.wt of T undecanoate (I); 1000 mg/kg b.wt of T undecanoate (J); 125 mg/kg b.wt of T propionate (K); 250 mg/kg b.wt of T propionate (L); 500 mg/kg b.wt of T propionate (M); 750 mg/kg b.wt of T propionate (N); 1000 mg/kg b.wt of T propionate (O). Epithelial thickness was measured (P). Both control groups and 1000mg/kg b.wt group consisted of 5 rats, other group consisted of 6 rats at the beginning of the experiment. Values are expressed as the mean ± SD. $^*p < 0.05$, $^{**}p < 0.01$, $^{***}p < 0.001$ compared with normal control (NC). Statistical analysis was performed by ANOVA and Tukey-Kramer multiple and Dunnett's multiple comparison test.

week maintained significantly high level of testosterone for 12 weeks, but injection intervals of 2–4 weeks showed unstable level of testosterone (Table 2). Compared with the NC group, DHT level in the CC group was significantly decreased (Table 2). At all injection intervals in T undecanoate-injected group significantly high level of DHT was maintained except high but not significant level at 8–10 weeks and no decrease in the level after 12 weeks (Table 2). In T propssionate-injected groups, significantly high level of DHT was maintained at injection interval of one week like T undecanoate; however, injection intervals of 2–4 weeks showed unstable level of DHT (Table 2). The level of 5-alpha reductase, which converts testosterone to DHT, was measured. Compared with the NC group, 5-alpha reductase level in the CC group was significantly decreased (Table 2). All interval of T undecanoate-injection, although 8 weeks and 10 weeks did not increase 5-alpha reductase level significantly, maintained high level of DHT and it did not decrease after 12 weeks (Table 2). In T propionate-injected groups, injection intervals of one week maintained significantly high level of 5-alpha reductase like T undecanoate but injection intervals of 2–4 weeks showed unstable level of 5-alpha reductase (Table 2).

**Changes in prostate measurements.** At the end of the experiment, changes in the prostate, which is the most important indicator of BPH, including weight and volume of the prostate, were measured. Compared with the NC group, all prostate measurements were decreased in the CC group, whereas administration of T undecanoate led to an increase beyond the values of the NC group; in particular, the intervals of two weeks and three weeks showed the highest levels (Table 3). Contrarily, in all T propionate-administered groups, a significant increase in prostate measurements was not observed (Table 3).

**Table 2. Changes in serum levels of testosterone, dihydrotestosterone, and 5-alpha reductase in the injection cycle substudy.**

| Testosterone (ng/ml) | | 0 week | 2 week | 4 week | 6 week | 8 week | 10 week | 12 week |
|---|---|---|---|---|---|---|---|---|
| NC | | 4.11±3.73 | 1.29±2.23 | 2.01±2.55 | 2.75±1.93 | 4.48±4.47 | 4.37±4.69 | 3.57±4.05 |
| CC | | 5.91±1.66 | 0.03±0.03 | 0.01±0.00 | 1.08±2.13 | 0.01±0.00 | 0.00±0.00 | 0.02±0.04* |
| T undecanoate | 1week | 2.94±2.12 | 11.94±0.74*** | 12.47±0.20*** | 12.63±0.08*** | 12.69±0.10** | 12.59±0.13** | 12.54±0.15*** |
| | 2week | 2.00±2.16 | 8.85±1.30*** | 11.50±0.76*** | 12.28±0.29*** | 12.22±0.32** | 12.22±0.13** | 12.20±0.13*** |
| | 3week | 3.26±1.24 | 10.58±1.46*** | 11.69±0.16*** | 10.27±3.00*** | 11.95±0.30** | 11.77±0.96** | 11.28±0.35*** |
| | 4week | 3.42±2.70 | 9.31±3.22*** | 9.57±2.02*** | 11.48±0.95*** | 11.31±0.38* | 11.28±0.43* | 10.63±0.77*** |
| T propionate | 1week | 5.11±0.93 | 10.59±1.53*** | 12.45±0.05*** | 12.24±0.36*** | 12.61±0.53** | 10.58±3.10* | 12.12±0.57*** |
| | 2week | 5.03±3.39 | 2.59±1.73 | 5.30±5.07 | 3.40±4.86 | 7..24±4.92 | 4.33±4.05 | 7.45±2.42* |
| | 3week | 6.27±1.64 | 1.81±2.71 | 10.35±2.29*** | 0.02±0.03 | 3.16±5.56 | 10.45±1.39* | 0.32±0.36 |
| | 4week | 3.36±0.62 | 2.24±2.86 | 0.01±0.00 | 0.86±1.69 | 0.01±0.00 | 3.06±5.52 | 1.27±2.54 |
| Dihydrotestosterone (ng/ml) | | | | | | | | |
| NC | | 1.92±0.6 | 1.08±0.34 | 1.68±0.52 | 1.46±0.33 | 1.85±0.55 | 1.80±0.80 | 1.34±0.92 |
| CC | | 2.23±0.25 | 0.63±0.31 | 0.42±0.10*** | 0.92±0.72 | 0.43±0.05** | 0.33±0.05*** | 0.50±0.26 |
| T undecanoate | 1week | 1.80±0.30 | 2.80±0.11*** | 2.83±0.03*** | 2.79±0.05** | 2.88±0.06* | 2.46±0.3 | 2.83±0.04*** |
| | 2week | 1.60±0.16 | 2.47±0.12*** | 2.69±0.09*** | 2.75±0.07** | 2.89±0.03* | 2.45±0.37 | 2.76±0.06*** |
| | 3week | 1.89±0.24 | 2.69±0.09*** | 2.76±0.03*** | 2.43±0.40* | 2.83±0.04 | 2.56±0.31 | 2.55±0.16** |
| | 4week | 1.86±0.42 | 2.56±0.34*** | 2.39±0.17* | 2.49±0.27* | 2.56±0.39 | 2.72±0.06* | 2.43±0.12** |
| T propionate | 1week | 2.15±0.13 | 2.67±0.13*** | 2.85±0.02*** | 2.80±0.05** | 2.90±0.03* | 2.68±0.3 | 2.82±0.15*** |
| | 2week | 2.07±0.58 | 1.67±0.47 | 1.87±0.71 | 1.44±0.88 | 2.08±1.02 | 1.69±0.83 | 2.25±0.27* |
| | 3week | 2.29±0.18 | 1.22±0.67 | 2.56±0.30** | 0.39±0.25* | 1.50±0.95 | 2.58±0.20 | 0.72±0.42 |
| | 4week | 1.99±0.15 | 1.38±0.76 | 0.44±0.09*** | 0.86±0.61 | 0.54±0.08** | 0.38±0.37*** | 0.65±0.81 |
| 5alpha-reductase (ng/ml) | | | | | | | | |
| NC | | 1.03±0.33 | 0.68±0.30 | 0.76±0.28 | 0.72±0.27 | 1.26±0.27 | 1.21±0.39 | 1.03±0.42 |
| CC | | 1.22±0.11 | 0.40±0.17 | 0.18±0.04** | 0.36±0.39 | 0.47±0.04*** | 0.28±0.06*** | 0.48±0.21* |
| T undecanoate | 1week | 1.02±0.21 | 1.54±0.08*** | 1.59±0.02*** | 1.59±0.01*** | 1.57±0.15 | 1.67±0.01 | 1.66±0.01* |
| | 2week | 0.89±0.19 | 1.32±0.10** | 1.51±0.05*** | 1.53±0.06*** | 1.67±0.02 | 1.65±0.02 | 1.64±0.03* |
| | 3week | 1.07±0.13 | 1.46±0.07*** | 1.54±0.05*** | 1.44±0.08** | 1.67±0.01 | 1.65±0.03 | 1.58±0.06* |
| | 4week | 1.08±0.25 | 1.40±0.19** | 1.38±0.16*** | 1.39±0.06** | 1.56±0.18 | 1.64±0.02 | 1.54±0.05* |
| T propionate | 1week | 1.17±0.10 | 1.51±0.10*** | 1.6±0.04*** | 1.55±0.01*** | 1.58±0.08 | 1.35±0.34 | 1.66±0.03* |
| | 2week | 1.16±0.28 | 0.96±0.17 | 0.99±0.47 | 0.69±0.53 | 1.30±0.53 | 1.16±0.47 | 1.42±0.14 |
| | 3week | 1.28±0.09 | 0.78±0.38 | 1.45±0.18*** | 0.14±0.08* | 0.91±0.29 | 1.60±0.06 | 0.71±0.35 |
| | 4week | 1.10±0.08 | 0.67±0.44 | 0.17±0.05*** | 0.33±0.36 | 0.53±0.08*** | 0.92±0.56 | 0.54±0.52 |

Changes in serum levels of testosterone, dihydrotestosterone, and 5-alpha reductase measured over a 12-week period after injecting testosterone undecanoate (T undecanoate) or testosterone propionate (T propionate). Castrated treatment groups were injected T undecanoate or T propionate at intervals of 1, 2, 3, or 4 weeks using the same concentration (125 mg/kg b.wt). Values are expressed as the mean ± SD (n = 4).

*p < 0.05

**p < 0.01

***p < 0.001 compared with normal control (NC). Statistical analysis was performed by ANOVA and Dunnett's multiple comparison test.

**Histomorphological changes in prostate tissue.** The change in the epithelial thickness of the prostate tissue, a characteristic of BPH, was measured by H&E staining. Compared with the NC group, the CC group showed a significant decrease, up to an unmeasurable level, in the thickness of the prostate tissue (Fig 4A and 4B). In the exogenous testosterone administration groups, the epithelial thickness of the prostate tissue significantly increased in the T undecanoate administration group at intervals of 1–3 weeks, with the highest increase at interval of 3 weeks, whereas it was not significantly increased in all T propionate administration group (Fig 4C–4K).

**Table 3. Prostate measurements in the injection cycle substudy.**

| | | Prostate weight (g) | Prostate volume (mm3) | Prostate index (%) | Injection number (times/ 12 weeks) |
|---|---|---|---|---|---|
| NC | | 1.33±0.13 | 3747.30±1692.39 | 0.31±0.05 | |
| CC | | 0.08±0.02*** | 259.11±89.57* | 0.02±0.01* | |
| T undecanoate | 1week | 1.68±0.28 | 7297.20±2731.69* | 0.48±0.08* | 12 |
| | 2week | 1.79±0.22 | 8239.62±2228.03** | 0.51±0.07* | 6 |
| | 3week | 1.85±0.28 | 7995.65±1399.95** | 0.51±0.11** | 4 |
| | 4week | 1.53±0.16 | 6450.75±1208.80 | 0.43±0.03 | 3 |
| T propionate | 1week | 1.54±0.21 | 4911.56±770.15 | 0.44±0.07 | 12 |
| | 2week | 1.65±0.11 | 6235.42±1213.58 | 0.46±0.03 | 6 |
| | 3week | 1.00±0.39 | 3158.10±800.08 | 0.25±0.09 | 3 |
| | 4week | 0.57±0.58 | 1970.33±1809.15 | 0.15±0.15* | 2 |

Prostate measurements after injection of testosterone propionate (T propionate) or testosterone undecanoate (T undecanoate). Castrated treatment groups were injected with T propionate or T undecanoate at intervals of 1, 2, 3, or 4 weeks, all using the same concentration (125 mg/kg). Prostate weight in grams was measured. Prostate index was calculated as (prostate weight/body weight) × 100. Prostate volume in cubic millimeters was measured using a caliper. Values are expressed as means ± SD (n = 4).

*p $<$ 0.05

**p $<$ 0.01

***p $<$ 0.001 compared with control. Statistical analysis was performed by ANOVA and Dunnett's multiple comparison test.

## Discussion

T undecanoate has been known to have the longest half-life among testosterone drugs currently used for the treatment of hypogonadism. To induce BPH, it is important to maintain a high level of androgen in the body for a long period using exogenous testosterone. We hypothesized that T undecanoate is more efficient to maintain stable and high androgen levels required to induce BPH than T propionate. Thus, we speculated the effectiveness of T undecanoate application in maintaining androgen levels necessary to induce BPH. This is the first study, to the best of our knowledge, on application of long-acting potential of T undecanoate to induce BPH instead to treat gonadal function by considering optimal dosage and dosing cycle.

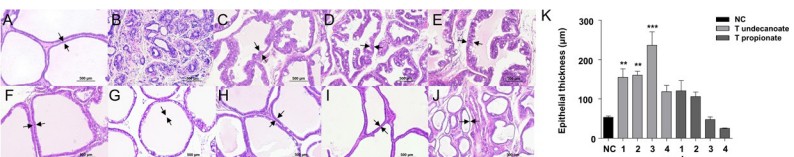

**Fig 4. Histomorphological changes in the prostate tissue.** Histomorphological changes in the prostate tissue. Castrated treatment groups were injected testosterone undecanoate (T undecanoate) or testosterone propionate (T propionate) at intervals of 1, 2, 3, or 4 weeks, all using the same concentration (125 mg/kg b.wt). Representative prostate tissue samples from each group were selected and stained with hematoxylin and eosin (H&E) and imaged at × 100 magnification. Microscopic images of the prostate samples (100× magnification) are shown from the following groups: Normal control (NC) (A); Castrated control (CC) (B); 1-week interval of T undecanoate (C); 2-week interval of T undecanoate (D); 3-week interval of T undecanoate (E); 4-week interval of T undecanoate (F); 1-week interval of T propionate (G); 2-week interval of T propionate (H); 3-week interval of T propionate (I); 4-week interval of T propionate (J). Epithelial thickness values are expressed as the mean ± SD (K). *p $<$ 0.05, **p $<$ 0.01, ***p $<$ 0.001 compared with normal control (NC). Statistical analysis was performed by ANOVA and Dunnett's multiple comparison test.

According to previous studies, factors that can indicate BPH include prostate weight, prostate volume, prostate index (prostate weight relative to body weight), prostate histopathology, and levels of testosterone, DHT, and 5-alpha reductase [18–27]. Prostatic hyperplasia is an obvious sign of BPH; in addition, DHT is the main hormone associated with prostatic hyperplasia, which is further related to the levels of testosterone and 5-alpha reductase, thus the three substances were considered as important indicators of BPH. Therefore, in this study, severity of prostatic hyperplasia was assessed using multiple factors, including prostate measurements, prostate histopathology, and serum levels of testosterone, DHT, and 5-alpha reductase.

We conducted two substudies to confirm optimal injection dosage and injection cycle for inducing BPH. The dose conditions of T undecanoate were selected upper 125mg/kg b.wt, which reported to maintain serum testosterone level not lower than control at least 4weeks [28]. In injection dosage substudy, exogenous testosterone was administered to castrated rats at different concentrations (125–1000 mg/kg b.wt). During the study, changes in hormone level was evaluated for eight weeks to determine the optimal dosage for injection cycle substudy.

Administration of T undecanoate showed dose-dependently increased levels of testosterone, DHT and in 5-alpha reductase; based on the changes in DHT-related markers, the dosage thought to be effective for repeated administration was reviewed. Additionally, we measured prostate measurements and epithelial thickness, which are major markers associated with enlargement of prostate tissue [29,30], to conform whether it affects the prostate level even after a long period of time after a single administration. The increasing epithelial thickness is well known feature of BPH [29–31], and it was confirmed concentration-dependently increases in exogenous testosterone administrated groups. In the group administered T undecanoate, a concentration-dependent decrease in the thymus weight was observed; the decrease in the thymus or spleen weight indicated that the treatment puts excessive stress on the body [35,36]. Therefore, 125 mg/kg b.wt T undecanoate was considered because it did not decrease prostate measurements due to castration compared with the NC group, even after eight weeks of single administration.

In injection cycle substudy, we administered exogenous testosterone to castrated rats at intervals of 1–4 weeks, using the concentration determined in the dosage substudy. T undecanoate stably maintained higher serum levels of testosterone, DHT, and 5-alpha reductase than T propionate. In the dosage substudy, 125 mg/kg b.wt T propionate did not maintain a significantly high level of testosterone two weeks after administration, and this result matched with 2–4 weeks of injection cycle using T propionate that could not maintain consistently high level of testosterone. Importantly, unlike administration of T propionate, which did not cause increase in the prostate measurements and epithelial thickness, the most important indicator for BPH, and even caused significant decrease as the injection cycle lengthened compared with the NC group, administration of T undecanoate at intervals of 1–3 weeks significantly increased the prostate measurements compared with the NC group.

In conclusion, based on the results of serum levels of testosterone, DHT, and 5-alpha reductase and prostate measurements, administration of 125 mg/kg b.wt T undecanoate at intervals of 2 or 3 weeks was confirmed as the optimal dosage and dosing cycle for stably and effectively inducing BPH, and this can reduce 14 to 21times of total injection number as replacing T propionate, which has used to induce BPH in previous studies, to T undecanoate [18–27,29–31]. In addition, through our two sub-studies may contribute to estimate the threshold at which the DHT level, which is significant for inducing BPH, is maintained according to the testosterone in the body after administration of exogenous testosterone.

This study, consisting of two substudies, showed the extended application of T undecanoate for induction of BPH, with advantages of maintenance that is more convenient and stable disease marker than inducing BPH using T propionate.

## Conclusions

We focused on the long-acting properties of T undecanoate for the maintenance of stable and high androgen levels over a long period and hypothesized that extended application of this feature would make it easier to induce BPH. We demonstrated extensive results on regulation of testosterone, DHT, and 5-alpha reductase levels in the body using various conditions of T undecanoate. Moreover, we utilized the long-acting potential of T undecanoate and showed that administration of 125 mg/kg b.wt T undecanoate every 2–3 weeks can induce BPH efficiently and stably. This study offers the possibility of extended application of T undecanoate to non-hypogonadism and may contribute to prostate research by providing an approach for inducing BPH efficiently and stably.

## Supporting information

**S1 Table. Organ weight and markers of liver and kidney in dosage substudy.** The levels of AST, ALT, Creatinine were measured by biochemical analyzer (AU480; Backman Coulter, Brea, CA, USA). Values are presented as means ± SD (n = 4). $^*$p < 0.05, $^{**}$p < 0.01, $^{***}$p < 0.001 compared with normal control (NC). Statistical analysis was performed by ANOVA and Dunnett's multiple comparison test.
(DOCX)

**S1 File. Dataset.**
(XLSX)

**S1 Graphical abstract.**
(TIF)

## Author Contributions

**Conceptualization:** Jinho An, Hyunseok Kong.

**Data curation:** Jinho An.

**Formal analysis:** Jinho An.

**Funding acquisition:** Hyunseok Kong.

**Methodology:** Jinho An.

**Project administration:** Hyunseok Kong.

**Supervision:** Hyunseok Kong.

**Writing – original draft:** Jinho An.

**Writing – review & editing:** Hyunseok Kong.

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
