## [Decision Letter · Decision Letter 0]

21 Feb 2022

PONE-D-21-40865Stable Hormone Regulation using Testosterone Undecanoate for Inducing Benign Prostatic Hyperplasia in Wistar RatPLOS ONE

Dear Dr. Kong,

Thank you for submitting your manuscript to PLOS ONE. After careful consideration, we feel that it has merit but does not fully meet PLOS ONE’s publication criteria as it currently stands. Therefore, we invite you to submit a revised version of the manuscript that addresses the points raised during the review process.

We look forward to receiving your revised manuscript.

Kind regards,

Yasmina Abd‐Elhakim

Academic Editor

PLOS ONE

Journal Requirements:

(This research was supported by a National Research Foundation of Korea grant funded by the Korean Government (MEST) (2019R1F1A1063286).)

(This research was supported by a National Research Foundation of Korea grant funded by the Korean Government (MEST) (2019R1F1A1063286).)

(HK

2019R1F1A1063286

National Research Foundation of Korea grant funded by the Korean Government

https://www.nrf.re.kr/index

The funders had no role in study design, data collection and analysis, decision to publish, or preparation of the manuscript.)

Reviewers' comments:

Reviewer's Responses to Questions

**Comments to the Author**

1. Is the manuscript technically sound, and do the data support the conclusions?

Reviewer #1: Yes

Reviewer #2: Yes

Reviewer #3: Yes

2. Has the statistical analysis been performed appropriately and rigorously? 

Reviewer #1: Yes

Reviewer #2: Yes

Reviewer #3: Yes

3. Have the authors made all data underlying the findings in their manuscript fully available?

Reviewer #1: Yes

Reviewer #2: Yes

Reviewer #3: No

4. Is the manuscript presented in an intelligible fashion and written in standard English?

Reviewer #1: Yes

Reviewer #2: Yes

Reviewer #3: Yes

5. Review Comments to the Author

Reviewer #1: Dear Authors,

Thank you for the opportunity to review your manuscript titled, "Stable Hormone Regulation using Testosterone Undecanoate for Inducing Benign Prostatic Hyperplasia in Wistar Rats ". The topic is interesting and of great importance. The authors determined the optimal dosage and dosing cycle using testosterone undecanoate for efficient and stable induction of benign prostatic hyperplasia than testosterone propionate. The manuscript is well written. There are certain issues that the authors need to address before the paper can be accepted for publication and I hope to be useful for improving the manuscript.

1- Please, titles and subtitles of manuscript parts should follow the instruction of the journal.

Title

2- The authors compared the using of testosterone undecanoate and / or testosterone propionate in induction of benign prostatic hyperplasia in all manuscript. So please mention this in your title. I suggest "Comparative application of testosterone undecanoate and / or testosterone propionate in induction of benign prostatic hyperplasia in Wistar rats"

Abstract

3- Please rewrite the method section determining the period of examining in both studies (8 and 12 weeks).

Introduction

4- Introduction is well written.

5- Please insert the full name beside the abbreviation if mentioned for first time, and please unite the abbreviations in all manuscript. Example. Testosterone undecanoate (T undecanoate), testosterone propionate (T propionate) and wrote in tables footnotes of table 3…(TU), (TP).

Materials and methods

6- Please insert feeding and drinking regime of rats during experiment period.

7- Please determine exact time of hormonal injection after castration.

8- Please insert dose of anesthetic agents used in castration of rats with reference.

9- The authors wrote "Each group consisted of 5–6 rats at the beginning of the experiment, total 69 rats were used" in dosage sub study section. Please determine the exact number of rats used in each group during dosage sub study and rewrite it in this study ' tables footnotes.

10- Please replace subtitle in line 112 " Androgens and 5-alpha reductase type 2" to "Blood sampling and hormonal analysis"

11- The authors wrote in statistical analysis section " Data are presented as the mean ± standard error of the mean (SEM)" and wrote in tables footnotes and figures legends "Standard deviation (SD)". Please check.

Results

12- Please use or remove the abbreviations in footnotes of table 3 concerning prostatic weight, volume and index in table 3.

13- Please insert arrows on figures of histopathology to clarifying the major differences between groups especially "epithelial thickness of the prostate tissue".

14- The authors not mentioned the parameters mentioned in supplementary (supporting) information except only in tables S1, S2 and in discussion section lines 290-292. Please check and add in details or remove.

Discussion

15- It needs more explanations for your results with needs of previous literatures that agree and disagree with your results.

Reviewer #2: This is the first study on application of long-acting potential of T undecanoate to induce BPH instead to treat gonadal function by considering optimal dosage and dosing cycle. It showed the extended application of T undecanoate for induction of BPH, with advantages of maintenance that is more convenient and stable disease marker than inducing BPH using T propionate.

Reviewer #3: Comments to the authors:

This manuscript entitled “Stable Hormone Regulation using Testosterone Undecanoate for Inducing Benign Prostatic Hyperplasia in Wistar Rats”. The manuscript is describing mainly the optimal dosage and dosing cycle for efficient and stable induction of benign prostatic hyperplasiausing using testosterone undecanoate. The injection cycle substudy, testosterone undecanoate-administered group stably maintained high levels of testosterone, dihydrotestosterone, and 5-alpha reductase compared with testosterone propionate-administered group for the same injection cycl. Some other points should be take in consideration for the improvement of the manuscript. The title needs much more modifications to match the designe. The manuscript is organized ,informative and well presented but needs some modifications.

Firistly , There were some notes

- The T undecanoate : should be mentioned in the first appearance as full term , then the abbreviation within the whole manuscript.

- Mention the most recent studies on prostatic hyperplasia causes , consequences and in-vivo models.

- The clinical uses and trials on exogenous testosterone should be updated and its clinical side effects.

- The rationale of selected doses for the treatment must be explained

- The attribution of the histopasthological alterations are not well described in the discussion , revise

- The conclusion needs to be rewritten in accordance to the aim and hypothesis of the study

6. PLOS authors have the option to publish the peer review history of their article (what does this mean?). If published, this will include your full peer review and any attached files.

Reviewer #1: No

Reviewer #2: **Yes: **qiling wang

Reviewer #3: No

---

## [Author Response · Author response to Decision Letter 0]

21 Mar 2022

This is our opinion in response to the reviewer’s comments on our manuscript, PONE-D-21-40865. In the revised manuscript, we had carefully examined our manuscript again and tried our best to clear the issues raised by the reviewers.

Reviewer #1: Dear Authors,

Thank you for the opportunity to review your manuscript titled, "Stable Hormone Regulation using Testosterone Undecanoate for Inducing Benign Prostatic Hyperplasia in Wistar Rats ". The topic is interesting and of great importance. The authors determined the optimal dosage and dosing cycle using testosterone undecanoate for efficient and stable induction of benign prostatic hyperplasia than testosterone propionate. The manuscript is well written. There are certain issues that the authors need to address before the paper can be accepted for publication and I hope to be useful for improving the manuscript.

1- Please, titles and subtitles of manuscript parts should follow the instruction of the journal.

Title

- We had carefully check again and it was modified. (line 1-2)

2- The authors compared the using of testosterone undecanoate and / or testosterone propionate in induction of benign prostatic hyperplasia in all manuscript. So please mention this in your title. I suggest "Comparative application of testosterone undecanoate and / or testosterone propionate in induction of benign prostatic hyperplasia in Wistar rats"

-We appreciate your suggestion.We changed the title following your suggestion. (line 1-2)

Abstract

3- Please rewrite the method section determining the period of examining in both studies (8 and 12 weeks).

- The period of examining in both studies was inserted in the method section. (line 22, 24)

Introduction

4- Introduction is well written.

5- Please insert the full name beside the abbreviation if mentioned for first time, and please unite the abbreviations in all manuscript. Example. Testosterone undecanoate (T undecanoate), testosterone propionate (T propionate) and wrote in tables footnotes of table 3…(TU), (TP).

Materials and methods

-The abbreviations in manuscreipt were checked and modified. (line 43, 51, 70, 246-248)

6- Please insert feeding and drinking regime of rats during experiment period.

- The feeding and drinking regime were inserted in method. (line 93)

7- Please determine exact time of hormonal injection after castration.

- We modified the procedure of horemanal injection in method. (line 93-95)

8- Please insert dose of anesthetic agents used in castration of rats with reference.

- The dose of anesthetic agents was inserted in method. (line 97)

9- The authors wrote "Each group consisted of 5–6 rats at the beginning of the experiment, total 69 rats were used" in dosage sub study section. Please determine the exact number of rats used in each group during dosage sub study and rewrite it in this study ' tables footnotes.

- The exact number of rats used in each group during dosage sub study rewrited in method and tables footnotes. (line 107-108)

10- Please replace subtitle in line 112 " Androgens and 5-alpha reductase type 2" to "Blood sampling and hormonal analysis"

- We changed the title following your suggestion. (line 116)

11- The authors wrote in statistical analysis section " Data are presented as the mean ± standard error of the mean (SEM)" and wrote in tables footnotes and figures legends "Standard deviation (SD)". Please check.

- Statistical analysis section was checked and modified. (line 141)

Results

12- Please use or remove the abbreviations in footnotes of table 3 concerning prostatic weight, volume and index in table 3.

- The abbreviations in footnotes of table prostatic weight, volume and index was removed. (line 248-249)

13- Please insert arrows on figures of histopathology to clarifying the major differences between groups especially "epithelial thickness of the prostate tissue".

-The arrows on figures of histopathology were inseted. (Figure 3, 4)

14- The authors not mentioned the parameters mentioned in supplementary (supporting) information except only in tables S1, S2 and in discussion section lines 290-292. Please check and add in details or remove.

- Explanation of the parameters in S1 was added in ditails and S2 was removed. (line 188-190)

Discussion

15- It needs more explanations for your results with needs of previous literatures that agree and disagree with your results.

- The explanation of comparison between our results and previous literatures was modified. (line 313-314)

Reviewer #2: This is the first study on application of long-acting potential of T undecanoate to induce BPH instead to treat gonadal function by considering optimal dosage and dosing cycle. It showed the extended application of T undecanoate for induction of BPH, with advantages of maintenance that is more convenient and stable disease marker than inducing BPH using T propionate.

-We appreciate your review.

Reviewer #3: Comments to the authors:

This manuscript entitled “Stable Hormone Regulation using Testosterone Undecanoate for Inducing Benign Prostatic Hyperplasia in Wistar Rats”. The manuscript is describing mainly the optimal dosage and dosing cycle for efficient and stable induction of benign prostatic hyperplasiausing using testosterone undecanoate. The injection cycle substudy, testosterone undecanoate-administered group stably maintained high levels of testosterone, dihydrotestosterone, and 5-alpha reductase compared with testosterone propionate-administered group for the same injection cycl. Some other points should be take in consideration for the improvement of the manuscript. The title needs much more modifications to match the designe. The manuscript is organized ,informative and well presented but needs some modifications.

- We appreciate your review and totally agree with your opinion. The title was modified to match the designe. (line 1-2)

Firistly , There were some notes

- The T undecanoate : should be mentioned in the first appearance as full term , then the abbreviation within the whole manuscript.

-The abbreviations in manuscreipt were checked and modified. (line 51)

- Mention the most recent studies on prostatic hyperplasia causes , consequences and in-vivo models.

-The reference of most recent studies on prostatic hyperplasia causes , consequences and in-vivo models was added. (line 55-58)

- The clinical uses and trials on exogenous testosterone should be updated and its clinical side effects.

- The clinical uses, trials and side effects on exogenous testosterone was updated (line 48-50)

- The rationale of selected doses for the treatment must be explained

- The explantaion of rationale of selected doses for the treatment was modified. (line 106)

- The attribution of the histopasthological alterations are not well described in the discussion , revise

- The attribution of the histopasthological alterations was added in discussion. (line 294-297)

- The conclusion needs to be rewritten in accordance to the aim and hypothesis of the study

- The conclusion was modified in accordance to the aim and hypothesis of the study. (line 322-324)

---

## [Decision Letter · Decision Letter 1]

1 Apr 2022

PONE-D-21-40865R1Comparative application of testosterone undecanoate and / or testosterone propionate in induction of benign prostatic hyperplasia in Wistar ratsPLOS ONE

Dear Dr. Kong,

Thank you for submitting your manuscript to PLOS ONE. After careful consideration, we feel that it has merit but does not fully meet PLOS ONE’s publication criteria as it currently stands. Therefore, we invite you to submit a revised version of the manuscript that addresses the points raised during the review process.

We look forward to receiving your revised manuscript.

Kind regards,

Yasmina Abd‐Elhakim

Academic Editor

PLOS ONE

Journal Requirements:

Reviewers' comments:

Reviewer's Responses to Questions

**Comments to the Author**

1. If the authors have adequately addressed your comments raised in a previous round of review and you feel that this manuscript is now acceptable for publication, you may indicate that here to bypass the “Comments to the Author” section, enter your conflict of interest statement in the “Confidential to Editor” section, and submit your "Accept" recommendation.

Reviewer #1: All comments have been addressed

Reviewer #2: All comments have been addressed

Reviewer #3: (No Response)

2. Is the manuscript technically sound, and do the data support the conclusions?

Reviewer #1: Yes

Reviewer #2: Yes

Reviewer #3: (No Response)

3. Has the statistical analysis been performed appropriately and rigorously? 

Reviewer #1: Yes

Reviewer #2: Yes

Reviewer #3: (No Response)

4. Have the authors made all data underlying the findings in their manuscript fully available?

Reviewer #1: Yes

Reviewer #2: Yes

Reviewer #3: (No Response)

5. Is the manuscript presented in an intelligible fashion and written in standard English?

Reviewer #1: Yes

Reviewer #2: Yes

Reviewer #3: (No Response)

6. Review Comments to the Author

Reviewer #1: Dear Authors,

Thank you for the opportunity to review your manuscript titled, "Stable Hormone Regulation using Testosterone Undecanoate for Inducing Benign Prostatic Hyperplasia in Wistar Rats ". The topic is interesting and of great importance. The authors determined the optimal dosage and dosing cycle using testosterone undecanoate for efficient and stable induction of benign prostatic hyperplasia than testosterone propionate. The manuscript is well written. There are few issues that the authors need to address and I hope to be useful for improving the manuscript.

1- Please, titles and subtitles of manuscript parts should follow the instruction of the journal.

Introduction

2- Please insert the full name beside the abbreviation if mentioned for first time, and please unite the abbreviations in all manuscript. Example BPH in page 3, line 49 and full name was mentioned in line 53. Also, WHO and FDA in lines 46, 47. Please check.

Results

The authors not mentioned the parameters mentioned in supplementary (supporting) information except only in tables S1. Please check and add in details or remove.

Reviewer #2: (No Response)

Reviewer #3: The authors addressed most of the required points but still major concerns about the rationale for the selected doses and also the hypothesis of the current study. In adddition the discussion still needs substantial revision especially regarding the results of histopathology especially of the prostate gland ( attributions and recent citations).

There is a nother major concern , Why the authors deleted the acknowlegment which is presented in the first copy of the manuscript. any concerns regarding this must be disclosed in a seperate file and disclose by all athors.

7. PLOS authors have the option to publish the peer review history of their article (what does this mean?). If published, this will include your full peer review and any attached files.

Reviewer #1: No

Reviewer #2: No

Reviewer #3: No

---

## [Author Response · Author response to Decision Letter 1]

21 Apr 2022

This is our opinion in response to the reviewer’s comments on our manuscript, PONE-D-21-40865. In the revised manuscript, we had carefully examined our manuscript again and tried our best to clear the issues raised by the reviewers.

Reviewer #1: Dear Authors,

Thank you for the opportunity to review your manuscript titled, "Stable Hormone Regulation using Testosterone Undecanoate for Inducing Benign Prostatic Hyperplasia in Wistar Rats ". The topic is interesting and of great importance. The authors determined the optimal dosage and dosing cycle using testosterone undecanoate for efficient and stable induction of benign prostatic hyperplasia than testosterone propionate. The manuscript is well written. There are few issues that the authors need to address and I hope to be useful for improving the manuscript.

1- Please, titles and subtitles of manuscript parts should follow the instruction of the journal.

- the instruction of the journal fo titles and subtitles of manuscript was check again.

Introduction

2- Please insert the full name beside the abbreviation if mentioned for first time, and please unite the abbreviations in all manuscript. Example BPH in page 3, line 49 and full name was mentioned in line 53. Also, WHO and FDA in lines 46, 47. Please check.

- The full name beside the abbreviation at the first time of mention was modified.(at line 46-47, 49, 54)

Results

The authors not mentioned the parameters mentioned in supplementary (supporting) information except only in tables S1. Please check and add in details or remove.

- The parameters of supplementary (supporting) information modified more detail.(at line 190-192)

Reviewer #3: The authors addressed most of the required points but still major concerns about the rationale for the selected doses and also the hypothesis of the current study. In adddition the discussion still needs substantial revision especially regarding the results of histopathology especially of the prostate gland ( attributions and recent citations).

- The discusstion of the rationale for the selected doses and the hypothesis of the current study were modified in detail. (at line 278-279, ) And, the discussions of the histopathology of the prostate gland and the hypothesis were modified. (at line 298-301)

There is a nother major concern , Why the authors deleted the acknowlegment which is presented in the first copy of the manuscript. any concerns regarding this must be disclosed in a seperate file and disclose by all athors.

- The acknowledgment, which presented in the first copy of the manuscript, was modified as directed by Journal Requirements (Journal Requirements: Please remove any funding-related text from the manuscript and let us know how you would like to update your Funding Statement.) in first revision. And, the acknowledgment content was included in the Funding Statement section of the online submission form.

---

## [Decision Letter · Decision Letter 2]

2 May 2022

PONE-D-21-40865R2Comparative application of testosterone undecanoate and / or testosterone propionate in induction of benign prostatic hyperplasia in Wistar ratsPLOS ONE

Dear Dr. Kong,

Thank you for submitting your manuscript to PLOS ONE. After careful consideration, we feel that it has merit but does not fully meet PLOS ONE’s publication criteria as it currently stands. Therefore, we invite you to submit a revised version of the manuscript that addresses the points raised during the review process.

We look forward to receiving your revised manuscript.

Kind regards,

Yasmina Abd‐Elhakim

Academic Editor

PLOS ONE

Journal Requirements:

**Reviewers' comments:**

**Reviewer #1: **Dear Authors,

Thank you for the opportunity to review again your manuscript titled, " Comparative application of testosterone undecanoate and / or testosterone propionate in induction of benign prostatic hyperplasia in Wistar rats " and thank you for response to previous reviewer comments.

• Please insert the methods of measuring of the parameters mentioned in supplementary (supporting) information.

**Reviewer #3:** (No Response)

---

## [Author Response · Author response to Decision Letter 2]

4 May 2022

This is our opinion in response to the reviewer’s comments on our manuscript, PONE-D-21-40865R2. In the revised manuscript, we had carefully examined our manuscript again and tried our best to clear the issues raised by the reviewers.

Journal Requirements:

-The reference list was carefully reviewed and modified (at line 339-341, 361, 363, 367, 371)

Reviewer #1

Please insert the methods of measuring of the parameters mentioned in supplementary (supporting) information.

- The methods of measuring of the parameters in supplementary (supporting) information was added (at line 419)

---

## [Editor Report · Decision Letter 3]

6 May 2022

Comparative application of testosterone undecanoate and / or testosterone propionate in induction of benign prostatic hyperplasia in Wistar rats

PONE-D-21-40865R3

Dear Dr. Kong,

We’re pleased to inform you that your manuscript has been judged scientifically suitable for publication and will be formally accepted for publication once it meets all outstanding technical requirements.

Kind regards,

Yasmina Abd‐Elhakim

Academic Editor

PLOS ONE
---

## [Editor Report · Acceptance letter]

10 May 2022

PONE-D-21-40865R3 

Comparative application of testosterone undecanoate and / or testosterone propionate in induction of benign prostatic hyperplasia in Wistar rats 

Dear Dr. Kong:

I'm pleased to inform you that your manuscript has been deemed suitable for publication in PLOS ONE. Congratulations! Your manuscript is now with our production department. 

Kind regards, 

on behalf of

Dr. Yasmina Abd‐Elhakim 

Academic Editor

PLOS ONE